# Effect of Nuts Combined with Energy Restriction on the Obesity Treatment: A Systematic Review and Meta-Analysis of Randomized Controlled Trials

**DOI:** 10.3390/foods13183008

**Published:** 2024-09-23

**Authors:** Darlene Larissa de Souza Vilela, Alessandra da Silva, Ana Claudia Pelissari Kravchychyn, Josefina Bressan, Helen Hermana Miranda Hermsdorff

**Affiliations:** 1Laboratory of Clinical Analysis and Genomics, Department of Nutrition and Health, Universidade Federal de Viçosa (UFV), Viçosa 36570-900, MG, Brazil; darlene.vilela@ufv.br (D.L.d.S.V.); ana.pelissari@ufv.br (A.C.P.K.); jbrm@ufv.br (J.B.); 2Laboratory of Energy Metabolism and Body Composition, Department of Nutrition and Health, Universidade Federal de Viçosa (UFV), Viçosa 36570-900, MG, Brazil; 3Laboratory of Health and Environment Education, Instituto Oswaldo Cruz, Rio de Janeiro 21040-360, RJ, Brazil; alessan.drasg94@gmail.com; 4Department of Epidemiology and Quantitative Methods in Health, Escola Nacional de Saúde Pública, Fundação Oswaldo Cruz, Rio de Janeiro 21040-900, RJ, Brazil

**Keywords:** energy restriction, overweight, obesity, oilseeds, weight loss, cardiometabolic diseases

## Abstract

Obesity is a multifactorial disease that is difficult to control worldwide. Although nuts are recognized health foods, the application of food in obesity management is unclear. We systematically reviewed the literature and performed a meta-analysis to evaluate if nut consumption favors people on energy restriction (ER) dietary interventions. Four databases were used to search for eligible articles in May 2024. This review was conducted according to the PRISMA guide, and the bias risk of papers was evaluated. For the meta-analysis, we extracted the endpoint values of the group’s variables and estimated the effect sizes by the random-effects model. Sixteen and ten articles were included in the systematic review and meta-analysis, respectively. Almonds were evaluated in the majority of studies (*n* = 6). The consumption of nuts (28 to 84 g/d, 4 to 72 months) included in ER (–250 to 1000 kcal/d) did not differently affect anthropometry (weight loss, BMI, waist and hip circumferences), body composition (fat mass, fat-free mass, or lean mass), markers of glucose (glycemia and insulinemia), lipid metabolism (total cholesterol, HDL-c, LDL-c, LDL-c/HDL-c, or triglycerides), and systolic and diastolic blood pressure. In most analyses, stratifying studies by type of nut or intervention time did not present different results in the meta-analysis. As there are few studies, in addition to great methodological variability, more high-quality trials are needed to confirm these results. The PROSPERO registration number is CRD42023444878.

## 1. Introduction

Obesity is a chronic multifactorial disease defined by excessive fat that can impair health. This disease continues to grow, and its prevalence has more than doubled since 1990 [1]. More than 2 billion adults had overweight in 2022 worldwide, of which 890 million were living with obesity. Indeed, high body mass index (BMI) values caused an estimated 5 million deaths from noncommunicable diseases such as cardiovascular diseases, diabetes, cancers, and others, which impair health and life expectancy [1].

Energy restriction (ER) is a consolidated strategy for weight loss [2,3,4,5,6]. However, ER has promoted moderate weight loss after 12 months [7], while it is difficult to maintain this weight loss in the long term [8]. Thus, the consumption of foods with additional beneficial effects on weight loss and improvement of cardiometabolic risk markers, regardless of the ER, such as functional foods and bioactive compounds derived from these foods, have been investigated [9,10].

In this sense, nuts (single-seeded dried fruits such as hazelnuts, chestnuts, and walnuts) are rich in bioactive compounds, high-quality proteins, fiber, minerals, tocopherols, phytosterols, and phenolic compounds [11,12]. Despite being rich in unsaturated fat, the consumption of nuts, up to 100 g/day, has not been associated with weight gain [13,14]. Some mechanisms may explain the potential beneficial effects of nut consumption in the context of an ER diet. Nuts can promote satiety, which would lead to a reduction in compensatory food intake, in addition to their low metabolizable energy and incomplete chewing, which generates energy loss through feces, in daily doses of 42 to 84 g [15]. Also, the capacity to decrease ghrelin concentrations can potentially reduce hunger in daily doses of 45 g (15 g of Brazil nuts + 30 g of cashew nuts) [16]. Nut intake (22.1 g to 56 g/day) could increase diet-induced thermogenesis and basal energy expenditure [17] and promote greater adherence to dietary therapy. Additionally, the dietary fiber from nuts (42 to 84 g/day) delays gastric emptying and subsequent absorption, which potentially suppresses hunger and promotes a healthy gut microbiome that improves energy metabolism, too [15]. Furthermore, the consumption of nuts (15 to 60 g/day), especially due its bioactive compounds, is also associated with the control of cardiometabolic disorders, such as diabetes, metabolic syndrome, and cardiovascular disorders [18,19,20].

Although it is controversial, nuts have been shown to promote weight reduction when they are consumed in the context of a habitual diet [21,22,23,24], and may even promote additional weight loss when consumed in an ER diet [25,26,27,28]; however, these studies have not yet been critically reviewed.

A recent scoping review discussed the effects of nuts combined with ER diets on weight, body composition, and glucose control [27]. However, to date, these potential effects have not been critically reviewed or meta-analyzed. Thus, this study systematically reviewed and performed a meta-analysis to evaluate whether the consumption of nuts favors weight loss, anthropometric and body composition variables, and traditional cardiometabolic risk factors during ER dietary treatment.

## 2. Materials and Methods

### 2.1. Protocol and Registration

This systematic review was written in accordance with the Preferred Reporting Items for Systematic Reviews and Meta-Analyses (PRISMA) guide [29], and the PRISMA checklist is available in Appendix A. The review protocol was registered in the International Prospective Register of Ongoing Systematic Reviews (PROSPERO), with a registration number (CRD42023444878), available at https://www.crd.york.ac.uk/prospero/display_record.php?RecordID=444878 (accessed on 16 August 2024).

### 2.2. Eligibility Criteria

Original randomized clinical trials carried out in adults and older adults (age >18 years) were considered eligible for review. The intervention should evaluate the intake of any type of nuts (e.g., peanuts, almonds, cashews, Brazil nuts, etc.) and different doses of nuts (e.g., 25, 45 g/d), associated with all ranges of daily ER (e.g., 250, 500, or 1000 kcal/d). Moreover, the primary outcome was the effect on weight loss, BMI, waist circumference (WC), and body composition (fat mass—FM, fat-free mass—FFM, or lean mass—LM, visceral adipose tissue—VAT, and sagittal abdominal diameter—SAD). The secondary outcomes were the cardiometabolic risk markers, as follows: systolic blood pressure (SBP), diastolic blood pressure (DPB), total cholesterol (TC), serum high-density lipoprotein cholesterol (HDL-c), low-density lipoprotein cholesterol (LDL-c), low-density lipoprotein cholesterol/high-density lipoprotein cholesterol ratio (LDL-c/HDL-c ratio), triglycerides (TGs), glucose, insulin, homeostatic model assessment for insulin resistance (HOMA-IR), and liver enzymes (alanine transaminase—ALT, aspartate transaminase—AST, gamma-glutamyl transferase—GGT, and alkaline phosphatase—ALP).

The criteria for non-inclusion or exclusion were (1) publications that did not constitute complete original studies, such as conference abstracts, case reports, letters to the editor, and literature reviews (narrative, integrative, systematic, and meta-analysis); (2) studies that did not evaluate the consumption of nuts associated with ER and/or weight loss; (3) studies with a stage of life other than adults and the elderly, and (4) animal model studies.

To identify the studies and formulate the central question, we used the acronym PICOS (P = population; I = intervention; C = comparison; O = outcome; S = study design), in which the central question was “Does the nut consumption associated with an energy-restricted diet favor weight loss and control in traditional cardiometabolic risk markers?” (Appendix A).

### 2.3. Search Strategy

The PubMed/MEDLINE, Embase, Cochrane/Central Register of Controlled Trials (CENTRAL), and Scopus databases were searched for eligible articles by two authors (D.L.S.V. and A.S.) in parallel and independently, in May 2024, without filters or restrictions on language and date of publication. The descriptors used were chosen according to Medical Subject Headings (MeSH), Descriptors in Health Sciences (DeSC), and Emtree terms in English. The Boolean operators OR and AND were used to associate one word with another in the search. The complete search strategy, by database, is presented in Appendix A. In addition to the aforementioned search, a reverse search was carried out in the references of the included articles to identify other potentially eligible studies.

The following indexed terms were used in the databases to search (through title, abstract, and keywords or all fields) studies reporting the relationship between nut consumption associated with energy restriction and its influence on weight: (Caloric Restriction OR Restriction, Caloric OR Calorie Restricted Diet OR Calorie Restricted Diets OR Diet, Calorie Restricted OR Restricted Diet, Calorie OR Caloric Restricted OR Restricted, Caloric OR Low-Calorie Diet OR Diet, Low-Calorie OR Low Calorie Diet OR Low-Calorie Diets) AND (Weight Loss OR Loss, Weight OR Losses, Weight OR Weight Losses OR Weight Reduction OR Reduction, Weight OR Reductions, Weight OR Weight Reductions) AND (Nuts OR Nut OR Sweet Almond OR Almonds OR Almond OR Brazil Nuts OR Brazil Nut OR Nut, Brazil OR Cashew OR Cashews OR Filberts OR Filbert OR Hazelnuts OR Hazelnut OR Macadamia OR Macadamias OR Macadamia Nut OR Macadamia Nuts OR Pecans OR Pecan OR Hickory Nuts OR Hickory Nut OR pine nuts OR Pistacia vera OR Pistachio OR Pistachios OR Walnut OR Walnuts OR English Walnuts OR Juglans nigra OR Black Walnut Arachis hypogaea OR Peanuts OR Peanut OR Baru nut OR tree nuts OR groundnut). The complete search for each database is available in Appendix A.

### 2.4. Data Extraction and Selection Process

Eligible articles were identified by screening titles, abstracts, and full texts, by two researchers independently and in parallel (D.L.S.V. and A.S.). Initially, search results from each database were imported into Rayyan QCRY^®^ software [30] to exclude duplicates.

The study selection process was carried out using the same software by two researchers (D.L.S.V. and A.S.) in a blind, independent, and parallel manner. In case of disagreements during the title evaluation process, the article was included in the next stage. In case of disagreements after complete reading, they were resolved by consensus between the two researchers (D.L.S.V. and A.S.) or by consultation with a third author (H.H.M.H).

After selecting the studies, a summary table was created. The aspects considered for the preparation of the summary table were decided by consensus between the authors and included reference (author and year of publication), study design and duration, sample characteristics, intervention characteristics, evaluated markers, and main results. Data were extracted by one reviewer (D.L.S.V.) using a standardized form in Microsoft^®^ Excel^®^ software 2019 version 16.0 with the variables of interest (Table 1) and extracted independently and checked for consistency by the second reviewer (A.S.). Discrepancies were resolved by consensus.

### 2.5. Assessment of Risk of Bias

All studies included in the systematic review were individually submitted to rigorous assessment for risk of bias by two researchers (D.L.S.V. and A.S.) in parallel and independently, and all discrepancies were resolved by consensus. When discrepancies persisted, a third author (H.H.M.H) was consulted. Critical analysis was based on the Joanna Briggs Institute (JBI) Reviewers’ Manual using Critical Analysis Tools specific to each study design, developed by the JBI and approved by the JBI Scientific Committee after extensive peer review [31]. The JBI manual states that authors must establish a priori the criteria used to define the level of bias for each study, as there is no established score to determine the level of risk of bias for each article (such as low, medium, or high) [30]. Therefore, a score was determined to individually classify the level of risk of bias for each article. According to the percentage of affirmative responses, articles were classified as having a low (≥70%), moderate (between 50 and 70%), and high risk of bias (<50%) [32].

### 2.6. Meta-Analysis

A meta-analysis of quantitative variables collected at the endpoint of the intervention period was performed to calculate the standard mean differences (SMDs) between treatment and control groups. Standard deviation was calculated for some studies with data of 95% confidence intervals and standard error. This way of meta-analyzing the data was chosen due to the lack of information about the mean difference (difference from the beginning to the end of the study) of the variables in most studies. To estimate effect sizes, the common-effects model and random-effects model were applied, and results were achieved based on the SMD and 95% confidence interval. The random-effects model was adopted considering the heterogeneity of interventions [33,34]. The *I*^2^ statistic was calculated to assess the heterogeneity (low: <50%; moderate to high: 50–75%; high: >75%). *p* Values < 0.05 were considered significant for all analyses [35]. Sensitivity analyses stratifying data by type of nuts and intervention time were performed. Due to the difficulty of obtaining information on the amount of energy restriction in some studies, the stratified analysis by the amount of energy restriction was not performed. The R software (meta package), version 4.3.3, was used for data synthesis.

## 3. Results

### 3.1. Study Selection and Characteristics

A total of 304 citations were retrieved from the PubMed/MEDLINE, Embase, Cochrane/Central, and Scopus databases from May 2024, and 827 from the reverse search. Of the citations retrieved from the databases, 223 titles remained after removing duplicates using Rayyan QCRY^®^ software. During title and abstract screening, 192 records were removed based on the initial exclusion criteria. Of the 31 titles requested to be read in full, 9 were not retained because they were conference abstracts. In the reverse search of the 827 records screened, 1 remained for complete reading. Finally, 22 articles were read in full, of which 6 were excluded (Figure 1). The reasons for excluding studies were as follows: the intervention did not include energy restriction (*n* = 3), the study only assessed satiety (*n* = 1), and body weight was assessed only at baseline (*n* = 2). Details of excluded papers are available in the Appendix A. At the end of the process, 16 articles were included in this systematic review (Figure 1). However, meta-analysis was considered only for the studies with endpoint values of markers for the control and nuts groups. Thus, ten studies were included in the majority of the analysis of the meta-analysis. Some variables were not evaluated in some studies or were presented in graphic format. Thus, in some cases, the meta-analysis comprised fewer studies than the systematic review.

This review included 16 randomized controlled trials, of which the majority included two arms (*n* = 13; 81.25%). Six studies were conducted in the USA [26,28,36,37,38,39], five were conducted in Iran [25,40,41,42,43], four were conducted in Brazil [44,45,46,47], and one was conducted in Australia [48]. The total initial sample consisted of 1.291 adults and elderly people aged ≥18 years, mainly women (67.31%; *n* = 869). The nuts used in the interventions were almonds (*n* = 6) [25,26,28,36,40], peanuts (*n* = 4) [44,45,46,48], pistachios (*n* = 2) [38,39], walnuts (*n* = 2) [37,41], a mix of nuts with equal amounts of pistachios, almonds, and peanuts (*n* = 2) [42,43], and another mix of nuts with 30 g of cashew nuts and 15 g of Brazil nuts (*n* = 1) [47]. Daily doses of nuts ranged from 28 to 84 g [28,36], with two studies not offering measurement in grams: one offered the daily energy value (15%) [26] and the other units (9 and 18 for each group) [41] Most studies (*n* = 14) used a nut-free energy-restricted diet as a control. The remaining two studies used a nut-free energy-restricted diet that was low in fat [48] and an energy-restricted diet with 300 g of fatty fish (salmon or trout)/week [41]. All studies had follow-up periods ranging from four [44,45] to 72 weeks [36]. Regarding energy restriction, one study offered approximately 1012 kcal/day [28], another study offered 1200–1500 kcal/day for women and 1500–1800 kcal/day for men [36], two others used energy requirements (REs)—25% (RE) [42,43], another presented values between −1300 to −1700 kcal/d [48], and a study used the total energy of diets: ~2000 kcal/day [41]. The majority of studies had an energy restriction that ranged from −250 [44,45,46] to −1000 kcal/day [25,37,40] (Table 1).

**Table 1 foods-13-03008-t001:** Characteristics of included studies that investigated the effect of nuts combined with energy restriction on the obesity treatment.

Reference	Study Design and Follow-Up	Sample Characteristics	Characteristics of the Intervention	Assessed Cardiometabolic Risk Markers	Main Results
Studies with Almonds					
Win et al., 2003 [28] USA	RCT, controlled 24 wk. 2 wk. run-in period	Individuals with obesity N: 65 */52 (CG = 33 and IG = 32) ** (F = 37; M = 28) Age: CG = 57 (SD 2) y. IG = 53 (SD 2) y. BMI: CG = 37 (SD 1) kg/m^2^ IG = 39 (SD 1) kg/m^2^	CG: nut-free CHO-energy-restricted formula-based diet (1015 kcal) IG: energy-restricted diet + 84 g/d of almonds (1012 kcal)	Anthropometry and body composition Glucose metabolism markers Lipid metabolism markers and blood pressure	IG: ↓ weight, BMI, WC, FM (kg), SBP, HDL-c vs. CG IG: ↔ FFM (kg), insulin, glucose, HOMA-IR, DBP, TC, TG, LDL-c, LDL-c/HDL-c ratio vs. CG
Foster et al., 2012 [36] USA	RCT, controlled 24 and 72 wk.	Individuals with overweight and obesity N: 123 */92 (CG = 62 and IG = 61) ** (F = 112; M = 11) Age: 46.8 (SD 12.4) y. BMI: 34.0 (SD 3.6) kg/m^2^	CG: nut-free energy-restricted diet IG: energy-restricted diet + 28 g of almonds/day Total energy of diets: 1200–1500 kcal/d for women and 1500–1800 kcal for men)	Anthropometry and body composition Glucose metabolism markers Lipid metabolism markers and blood pressure	CG: ↓ weight at 24 wk vs. IG IG: ↓ TC, TG, TC/HDL-c at 24 wk. vs. CG IG: ↔ weight, TG, TC, TC/HDL-c ratio at 72 wk. vs. CG IG: ↔VLDL, LDL-c, HDL-c, SBP, DBP, LM, and FM at 24 and 72 wk. vs. CG
Abazarfard; Salehi; Keshavarzi, 2014 [25] Iran	RCT, controlled 12 wk.	Premenopausal women with overweight N: 108 */100 (CG = 50 and IG = 50) ** Age: CG= 42.94 (SD 6.82) y. IG = 42.36 (SD 7.30) y. BMI: CG = 29.37 (SD 1.73) kg/m^2^ IG = 29.91 (SD 1.20)	CG: nut-free energy-restricted diet IG: energy-restricted diet + 50 g of raw almonds/d Energy-restricted diet for both groups: −1000 kcal	Anthropometry Glucose metabolism markers Lipid metabolism markers and blood pressure	IG: ↓ weight, BMI, WC, WHR, TG, TC, TC/HDL-c ratio, glucose, DBP vs. CG CG: ↓ HC, LDL-c, SBP, and ↑ HDL-c vs. IG
Abazarfard et al., 2016 [40] Iran	RCT, controlled 12 wk.	Premenopausal women with overweight N: 108 */100 (CG = 50 and IG = 50) ** Age: CG = 42.94 (SD 6.82) y. IG = 42.36 (SD 7.30) y. BMI: CG = 29.37 (SD 1.73) kg/m^2^ IG = 29.91 (SD 1.20)	CG: nut-free energy-restricted diet IG: energy-restricted diet + 50 g of raw almonds/d Energy-restricted diet for both groups: −1000 kcal	Liver enzymes	IG: ↓ ALT, AST, and GGT vs. CG IG: ↔ ALP, total bilirubin, albumin, and total protein vs. CG
Dhillon; Tan; Mattes, 2016 [26] USA	RCT, controlled 12 wk.	Individuals with overweight and obesity N: 86 */79 (CG = 27 and IG = 23) ** (F = 65; M = 21) Age: CG = 34.9 (SD 13.1) y. IG = 33.6 (SD 12.9) y. BMI: CG: 30.6 (SD 3.9) kg/m^2^ IG: 30.3 (SD 3.2) kg/m^2^	CG: nut-free energy-restricted diet IG: energy-restricted diet + 15% of energy from almonds/d Energy-restricted diet for both groups: −500 kcal	Anthropometry Glucose metabolism markers Lipid metabolism markers and blood pressure	IG: ↓ DBP, trunk FM (kg and %), FM (%) vs. CG IG: ↑ FFM (%) and truncal FFM (%) vs. CG IG: ↔ weight, VAT, SBP WC, SAD, insulin, glucose, TG, TC, LDL-c, HDL-c, FT (kg), FFM (kg), and truncal FFM (kg) vs. CG
Studies with Peanuts					
Alves et al., 2014 [44] Brazil	RCT, controlled 4 wk.	Men with overweight N: 76 */65 (CG = 22, IG1 = 22, and IG2 = 21) ** Age: CG = 27.4 (SEM 1.6) y. IG1 = 28.0 (SEM 1.5) y. IG2 = 26.8 (SEM 1.9) y. BMI: CG = 29.7 (SEM 0.53) kg/m^2^ IG1 = 29.5 (SEM 0.4) kg/m^2^ IG2 = 29.9 (SEM 0.6) kg/m^2^	CG: nut-free energy-restricted diet IG1: energy-restricted diet + 56 g of conventional peanuts IG2: energy-restricted diet + 56 g of high-oleic peanuts Energy-restricted diet for all groups: −250 kcal/d	Anthropometry and body composition	IG1 and IG2: ↔ weight, BMI, WC, HC, WHR, FM, FFM, LM vs. CG
De Oliveira Fialho et al., 2022 [46] Brazil	RCT, controlled 8 wk.	Premenopausal women with obesity N: 26 */24 (CG = 8, IG1 = 8, and IG2 = 8) ** Age: 33.1 (SD 8.7) y. BMI: 34.3 (SD 3.7) kg/m^2^	CG: nut-free energy-restricted diet IG1: energy-restricted diet + 56 g of whole roasted peanuts IG2: energy-restricted diet + 56 g of skinless peanuts Energy-restricted diet for all groups: −250 kcal/d	Anthropometry and body composition Glucose metabolism markers Lipid metabolism markers Liver enzymes	Week 4 IG1: ↔ weight, WC, WHR, LM (%), FM (%), SBP, DBP, TC, HDL-c, VLDL, LDL-c, LDL-c/HDL-c ratio, TG, glucose, insulin, HOMA-IR, AST, and ALT CG IG1: ↓ BMI vs. CG IG2: ↑ weight, WC, FM (%) vs. CG IG2: ↔ BMI, WHR, SBP, DBP, TC, HDL-c, VLDL, LDL-c, LDL-c/HDL-c ratio, TG, glucose, insulin, HOMA-IR, AST, and ALT vs. CG IG2: ↓ LM (%) vs. CG Week 8 IG1: ↔ weight, WC, WHR, LM (%), FM (%), SBP, DBP, HDL-c, VLDL, LDL-c, LDL-c/HDL-c ratio, TG, glucose, insulin, HOMA-IR, AST, and ALT vs. CG IG1: ↓ BMI, TC vs. CG IG2: ↑ weight, WC, FM (%) vs. CG IG2: ↔ BMI, WHR, SBP, DBP, HDL-c, VLDL, LDL-c, LDL-c/HDL-c ratio, TG, glucose, insulin, HOMA-IR, AST, and ALT vs. CG IG2: ↓ LM (%) and TC vs. CG
Petersen et al., 2022 [48] Australia	RCT, controlled 12 and 24 wk.	Individuals with overweight or obesity and high risk for T2DM N: 107 */76 (CG = 32 and IG = 44) ** (F = 70; M = 37) Age: 58 (SD 14) y. BMI: 33.1 (SD 5.4) kg/m^2^	CG: nut-free energy-restricted diet and low in fat: energy-restricted diet + 70 g of peanuts Energy restriction in both groups: −1300 and −1700 kcal/d)	Anthropometry Glucose metabolism markers Blood pressure	Week 12 IG: ? weight, SBP, DBP, glucose, insulin, HOMA-IR vs. CG IG: ↔ 2-h-glucose vs. CG Week 24 IG: ↔ weight vs. CG IG: ↓ SBP vs. CG IG: ? DBP, glucose, insulin, HOMA-IR vs. CG
Caldas et al., 2020 [45] Brazil	RCT, controlled 4 wk.	Men with overweight N: 76 */64 ** Age: 27.53 (SD 0.9) y. BMI: 29.76 (SD 0.3) kg/m^2^	CG: nut-free energy-restricted diet IG1: energy-restricted diet + 56 g of conventional peanuts IG2: energy-restricted diet + 56 g of high-oleic peanuts Energy-restricted diet for all groups: −250 kcal/d	Oxidative stress Inflammation	CG: ↔ IL-17A, IL-10, IL-6, IL-4, TNF, CRP, GST, ON, and SOD vs. IG1 and IG2 CG: ↓ MDA vs. IG and IG2
Studies with Walnuts					
Rock et al., 2017 [37] USA	RCT, controlled 12 and 24 wk.	Individuals with overweight and obesity without diabetes N: 100 * (CG = 51 and IG = 49) (F = 58; M = 42) Age: CG = 52.2 (SE 1.6) y. IG = 53.3 (SE 1.4) y. BMI: CG = 32.4 (SE 0.4) kg/m^2^ IG = 32.4 (SE 0.5) kg/m^2^	CG: nut-free energy-restricted diet IG: energy-restricted diet + 42 g of walnuts/d for diet ≥ 1500 kcal/d or 28 g for diet < 1500 kcal/d (~15% TEV) Energy-restricted diet for both groups: −500 to −1000 kcal/d	Anthropometry and body composition Glucose metabolism markers	IG: ↔ weight, BMI, WC, SBP, DBP, TG, HDL-c, and TC vs. at 12 and 24 wk. CG
Fatahi et al., 2019 [41] Iran	RCT, controlled 12 wk.	Postmenopausal women with overweight and obesity N: 66 * (CG = 33, IG = 33) Age: 53.5 (SD 1.6) y. BMI: 33.29 (SD 5.63) kg/m^2^	CG: energy-restricted diet + 300 g of fatty fish (salmon or trout)/week IG: energy-restricted diet + 18 units of walnuts/week and avoid fish intake Total energy of diets: ~2000 kcal/d	Anthropometry Glucose levels Liver enzymes Blood pressure Oxidative stress and inflammation markers	IG: ↔ weight, WC, MDA, AST, vs. CG IG: ↓ DBP, ↑ HDL-c vs. CG CG: ↓ SBP, glucose, TG, LDL-c, hs-PCR, ALT, TNF, IL-6 vs. IG
Studies with Pistachios					
Li et al., 2010 [38] USA	RCT, controlled 6 and 12 wk.	Individuals with overweight and obesity N: 70 */52 ** (CG = 25 and IG = 27) (F = 57; M = 13) Age: CG = 47.30 (SD 2.3) y. IG = 45.40 (SD 2.0) y. BMI: CG = 30.9 (SE 0.4) kg/m^2^ IG = 30.1 (SE 0.4) kg/m^2^	CG: nut-free energy-restricted diet +56 g of pretzels IG: energy-restricted diet + 53 g of pistachios Energy-restricted diet for all groups: −500 kcal/d	Anthropometry Glucose metabolism markers Lipid metabolism markers	IG: ↓ BMI and TG at 6 and 12 wk. vs. CG IG: ↔ weight, TC, LDL-c, HDL-c, glucose, and insulin at 6 and 12 wk. vs. CG
Rock et al., 2020 [39] USA	RCT, controlled 16 wk.	Individuals with overweight or obesity without diabetes N: 100 */93 ** (CG = 47 and IG = 49) (F = 62; M = 38) Age: CG = 56.2 (SE 1.5) y. IG = 55.0 (SD 1.6) y. BMI: CG = 32.8 (SE 0.5) kg/m^2^ IG = 32.8 (SD 0.6) kg/m^2^	CG: nut-free energy-restricted diet IG: energy-restricted diet + 42 g of pistachios (18% of TEV) Energy-restricted diet for all groups: −500 to −1000 kcal/d	Anthropometry Glucose metabolism markers Lipid metabolism markers Blood pressure	IG: ↔ weight, WC, SBP, DBP, TC, TG, HDL-c, LDL-c, insulin, glucose, and HOMA-IR vs. CG
Studies with Mixed Nuts					
Ghanavati et al., 2021 (A) [42] Iran	RCT, controlled 8 wk.	Individuals with overweight and obesity and stable coronary artery disease N: 70 */67 ** CG = 32 and IG = 35) (F = 30; M = 37) Age: 58.86 (SD 7.47) y. BMI: 30.9 (SD 3.9) kg/m^2^	CG: nut-free energy-restricted diet IG: energy-restricted diet + 20% energy from a mix of nuts (39 to 60 g with equal amounts of pistachios, almonds, and peanuts) Energy-restricted diet for both groups: −25% of RE	Anthropometry Inflammation	IG: ↓ ICAM-1 and IL-6 vs. CG IG: ↔ weight, BMI, WC, CRP, IL-10, and MCP-1 vs. CG
Ghanavati; Ali-Pour Parsa; Nasrollahzadeh, 2021 (B) [43] Iran	RCT, controlled 8 wk.	Individuals with overweight and obesity and stable coronary artery disease N: 70 */67 ** CG = 32 and IG = 35) (F = 30; M = 37) Age: 58.86 (SD 7.47) y. BMI: 30.9 (SD 3.9) kg/m^2^	CG: nut-free energy-restricted diet IG: energy-restricted diet + 20% energy from a mix of nuts (39 to 60 g with equal amounts of pistachios, almonds, and peanuts) Energy-restricted diet for both groups: −25% of RE	Anthropometry and body composition Glucose metabolism markers Lipid metabolism markers Liver enzymes	IG: ↑ HDL-c and Apo A1 vs. CG IG: ↔ weight, FM (%), MM (kg), TG, TC, LDL-c, sdLDL-c, TC/HDL-c ratio, LDL-c/HDL-c ratio, non-HDL-c, ALT, AST, and uric acid vs. CG IG: ↔ ABCA1 and ABCG1 gene expression vs. CG
Caldas et al., 2022 [47] Brazil	RCT, controlled 8 wk. 7–10 days of run-in	Women with overweight + at least one marker of metabolic syndrome or obesity regardless of the presence of metabolic syndrome components N: 40 */29 ** (CG = 15 and IG = 14) Age: 31.4 (SD 2.4) y BMI: 33.4 (SD 1.1) kg/m^2^	CG: nut-free energy-restricted diet IG: energy-restricted diet + 45 g of mixed nuts (30 g of cashew nuts + 15 g of Brazil nuts) Energy-restricted diet for all groups: −500 kcal/d	Anthropometry and body composition Glucose metabolism markers Lipid metabolism markers	IG: ↓ FM, VCAM-1 vs. CG IG: ↑ LM (%), FFM (%), MMT, TLM, AFM, and blood selenium levels vs. CG IG: ↔ weight, BMI, WC, WHR, HC, NC, WHtR, FM (kg), LM (kg), FFM (KG), TC, TG, LDL-c, HDL-c, VLDL, non-HDL-c, TC/HDL-c ratio, LDL-c/HDL-c ratio, glucose, insulin, TyG index, Apo A1, Apo B, Apo E, ApoB/Apo A, hs-CRP, DBP, SBP, ABI, NO, and ICAM-1 vs. CG

Legend: * initial *n* (randomized individuals); ** final *n* (accounting for losses to follow-up); ↑ increased; ↓ decreased; ↔ unchanged; ? without information; y: years; SD: standard deviation; F: female; M: male; BMI: body mass index; CG, control group; IG intervention group; LCD: low-calorie diet; WC: waist circumference; BF: body fat; TBW: total body water; SBP: systolic blood pressure; FFM: fat-free mass; HOMA-IR: homeostatic model assessment; DBP: diastolic blood pressure; TC: total cholesterol; TG: triglyceride; LDL-c: low-density lipoprotein cholesterol; HDL: high-density lipoprotein cholesterol; RMR: resting metabolic rate; FPG: fasting plasma glucose; AED: almond-enriched diet; NFD: nut-free diet; LM: lean mass; VLD-C: very low-density lipoprotein cholesterol; WHR: waist/hip ratio; RE: energy requirement; CTRL: control; CVP: conventional peanut; HOP: high-oleic peanut; RQ: respiratory quotient; DIT: diet-induced thermogenesis; ALT: alanine aminotransferase; AST: aspartate aminotransferase; GGT: gamma-glutamyl transferase; ALP: alkaline phosphatase; MGT: truncal fat mass; TFFM: truncal fat-free mass; CRP-US: C-reactive protein—ultra sensitive; TNF-α: tumor necrosis factor alpha; IL-6: Interleukin 6; MDA: Malondialdehyde; WP: whole peanut; SP: skinned peanut; NP: no peanut; ox-LDL-c: oxidized low-density lipoprotein; TAC: Total Plasma Antioxidant Capacity; NELCD: nut-enriched low-calorie diet; sdLDL: small dense LDL-c; Apo A1: apolipoprotein A-1; ABCA1: ATP-binding cassette transporter A1; ABCG1: ATP-binding cassette subfamily G member 1; NFLCD: nut-free low-calorie diet; ICAM-1: intercellular adhesion molecule-1; IL-10: Interleukin 10; MCP-1: monocyte chemotactic protein-1; BN: Brazil nut; WHtR: waist-to-height ratio; VCAM-1: Vascular Cell Adhesion Molecule; TLM: Truncal Lean Mass; AFM: Android Fat Mass; HbA1c: Glycated Hemoglobin; ΔAUC: Delta Area Under the Curve; GIP: gastric inhibitory peptide; GLP-1: Glucagon-like Peptide 1.

### 3.2. Outcomes by Type of Nuts

#### 3.2.1. Almonds

In total, five studies analyzed the effect of almond consumption combined with ER. Wien and colleagues (2003) showed that 84 g/d of almond consumption within the ER diet (−1000 kcal/d for patient with overweight) was able to reduce weight, BMI, WC, FM, SBP, and HDL-c of individuals with obesity compared to a nut-free ER diet for 24 weeks. On the other hand, changes in FFM, glucose and lipid metabolism markers, and DBP did not differ between groups [28]. The intake of 50 g/d of almonds with an ER (−1000 kcal/d) diet for 12 weeks reduced weight, BMI, WC, WHR, as well as some lipid metabolism markers, glucose, and DBP in premenopausal and sedentary females, compared to a control group of premenopausal women with overweight. On the other hand, the control group reduced HP, LDL-c, and SBP, while increasing HDL-c, compared to the almonds group [25]. In the same intervention, in another study, the almonds group reduced liver enzymes (ALT, AST, and GGT) compared to the control group, with no differences for alkaline phosphatase (ALP), total bilirubin, albumin, and total protein between groups [40].

Contrary to these studies, the consumption of 15% of almonds of the total energy value allied to an ER diet (−500 kcal/d) did not reduce weight, VAT, WC, SBP, SAD, and glucose and lipid metabolism markers in individuals with overweight and obesity compared to the control group. Despite this, almond consumption decreased trunk and total FM, while it increased truncal and total FFM compared to the control group [26]. In the study by Foster and colleagues (2012), a 28 g/d of almond intake with a diet providing 1200–1500 kcal/d for women and 1500–1800 kcal/d for men decreased TC, TG, and the TC/HDL-c ratio at 24 weeks in individuals with BMIs of 27–40 kg/m^2^, compared with the control group, but not at 72 weeks. However, the control group had a more pronounced weight loss at 24 weeks, with no difference from the almond group at 72 weeks’ intervention. In addition, markers such as VLDL, LDL-c, HDL-c, SBP, DBP, LM, and FM did not differ between groups at 24 and 72 weeks of intervention [36].

In summary, the inclusion of almonds in an ER diet decreased weight in two out of four studies compared to the control groups, while the control group reduced weight compared to the nuts group in one study. For other markers, the studies’ data regarding almonds showed controversial results regarding their health benefits when compared to the control group. Different doses of almonds and times of intervention were observed, which could justify the inconclusive results.

#### 3.2.2. Peanuts

Four studies that were included in our analyses examined peanut consumption in addition to ER. In men with BMIs between 26 and 35 kg, the intake of 56 g/d of conventional or high-oleic peanuts allied to an ER diet (−250 kcal/d) during 4 weeks had no effects on weight, BMI, WC, HP, WHR, FM, FFM, and LM when compared to the control group [44]. In the study by Caldas and collaborators, with the intake of an ER diet (−250 kcal/d), there was no difference in adiposity indicators (BMI, WC weight) when the consumption of 56 g of high-oleic or conventional peanuts was compared to a peanut-free diet [45].

In another study, the consumption of 56 g/d of whole roasted peanuts decreased BMIs at 4 and 8 weeks and TC at 8 weeks, while it did not affect weight, WC, WHR, LM, FM, SBP, DBP, HDL-c, VLDL, LDL-c, LDL-c/HDL-c ratio, TG, glucose, insulin, HOMA-IR, AST, and ALT compared to the control peanut-free diet. In contrast, the consumption of 56 g/d of skinless peanuts plus an ER (–250 kcal/d) diet increased weight, WC, and FM while it decreased LM at 4 and 8 weeks of intervention compared to the control group [46]. In accordance with these results, Petersen and colleagues (2022) showed that 70 g/d of peanuts allied to a 1300 kcal/d diet for women and 1700 kcal/d for men during 24 weeks had no effect on weight but decreased SBP in individuals with overweight or obesity and a high risk for type 2 diabetes mellitus compared to the control. The missing comparison of glucose metabolism markers and blood pressure did not permit to us to infer differences between groups [48].

#### 3.2.3. Walnuts

Two studies that evaluated the effect of walnuts plus ER were included. The consumption of 28 to 42 g/d of walnuts with an ER diet (–500 to –1000 kcal/d) during 24 weeks did not affect weight, BMI, WC, SBP, DBP, TG, HDL-c, and TC compared to the control in individuals with overweight and obesity without diabetes [37]. Fatahi and colleagues (2019) studied available postmenopausal women with overweight or obesity who consumed 18 units of walnuts per week plus an ER diet for 12 weeks. The authors observed a decreased DBP and increased HDL-c with unchanged weight, WC, and AST compared to a control group that received an ER diet plus 300 g of fatty fish per week [41].

In summary, the studies investigating the walnut effects plus an ER diet on health metabolism markers are limited and controversial. We also highlight the limitation of Fatahi and colleagues (2019), who used fatty fish as a control.

#### 3.2.4. Pistachios

Two studies with pistachios were included in this systematic review. The intake of 53 g/d of pistachio associated with an ER diet (−500 kcal) decreased BMI and TG, compared to the control group at 6 and 12 weeks of intervention in individuals with overweight and obesity. In parallel, the consumption of pistachios in the context of ER did not affect weight, TC, LDL-c, HDL-c, glucose, and insulin compared to the control at both periods [38]. Another study investigated the eating of 42 g/d of pistachio with an ER diet (−500 to −1000 kcal/d) did not affect weight, WC, SBP, DBP, TC, TG, HDL-c, LDL-c, insulin, glucose, and HOMA-IR compared to the control group during 16 weeks of intervention [39].

#### 3.2.5. Mixed Nuts

Three studies developed their interventions with a nut mix. Eating 20% of daily energy from mixed nuts (pistachios + almonds + peanuts) plus an ER diet (75% of the estimated total energy expenditure) increased HDL-c, compared to the control, at 8 weeks in individuals with overweight and obesity and stable coronary artery disease. However, this same intervention, in another study, had no effects on weight, BMI, WC, FM, MM, TG, TC, LDL-c, small-density LDL-c, TC/HDL-c ratio, LDL-c/HDL-c ratio, non-HDL-c, ALT, and AST, compared to the control group in patients with obesity and stable coronary artery disease [42,43].

The eating of 45 g/d of mixed nuts (30 g of cashew + 15 g of Brazil nuts) plus ER (–500 kcal/d) during 8 weeks decreased FM and VCAM-1 and, in parallel, increased LM, FFM, MMT, truncal LM, and truncal FFM, and selenium, compared to the control group in adult women with overweight and cardiometabolic risk [47]. Despite this, no differences were observed between groups for markers of lipid and glucose metabolism.

The studies with mixed nuts are few and differ greatly in the type of nuts and sample characteristics. At present, translational beneficial outcomes are related to improving body composition (most preserved LM and FFM) and traditional cardiovascular markers. The sample characteristics, the summary of the interventions, and their main results are presented in Table 1**.**

### 3.3. Results of Risk of Bias Assessment

Appendix A summarizes the results of risk of bias assessment. Most studies showed sufficient information and had a moderate (69%; *n* = 11) risk of bias. Only 25% (*n* = 4) of studies showed a high risk of bias, and one study showed a low risk of bias (6%; *n* = 1). Intention-to-treat (ITT) analysis was mentioned in six articles, which justifies the increased risk prevalence of bias in the topic “individuals analyzed in the groups to which they were randomized”. Most studies did not mention the number and training of evaluators and their intra and inter reliability, which give the largest number of “unclear” answers. They also did not describe whether outcome assessors were blind to the treatment assigned. Half of the studies did not provide sufficient data on randomization to assess if the allocation was blind, creating bias in this aspect. Furthermore, two components, one relating to blinding the participant to the designated intervention and the researcher, received “no” answers; however, these items do not necessarily indicate bias, as due to the nature of studies involving dietary interventions, this blinding is not possible (Appendix A and Appendix A).

### 3.4. Meta-Analysis Results

Results of the baseline groups are in the Appendix A. For this topic, we bring the results of studies separated by nut type and cardiometabolic risk factors.

#### 3.4.1. Effects of Nut Consumption on Anthropometric Measurements and Body Composition Indicators during Energy-Restricted Dietary Intervention

The consumption of nuts allied to an ER diet had no effects on body weight (SMD: −0.18; 95% CI: −0.40 to 0.03), BMI (SMD: −0.45; 95% CI: −0.90 to 0.00), WC (SMD: −0.04; 95% CI: −0.36 to 0.28), HC (SMD: 0.05; 95% CI: −0.45 to 0.54), FM (SMD: 0.11; 95% CI: −0.43 to 0.64), LM or FFM (SMD: −0.02; 95% CI: −0.38 to 0.33) compared to controls. Heterogeneity (*I*^2^*)* was equal to 51%, 83%, 76%, 69%, 72%, and 59% for body weight, BMI, WC, HP, FM, and LM, respectively, suggesting moderate to high heterogeneity between studies (Figure 2, Figure 3, Figure 4, Figure 5, Figure 6 and Figure 7). Almond intake decreased WC compared to controls (SMD: −0.93; 95% CI: −1.41 to −0.45) with moderate heterogeneity of evidence (*I*^2^ = 53%) (Appendix A). In other subgroup analyses by nut type and time of intervention, eating nuts associated with an ER diet had no effects on anthropometric and body composition variables (Supplementary Appendix A).

#### 3.4.2. Effects of Nut Consumption on Lipid Metabolism Markers during Energy-Restricted Dietary Intervention

In general analysis, nut intake allied to an ER diet had no effects on TC (SMD: −0.26; 95% CI: −0.60 to 0.07), LDL-c (SMD: 0,09; 95% CI: −0.08 to 0.25), HDL-c (SMD: 0.07; 95% CI: −0.12 to 0.27), and LDL-c/HDL-c ratio (SMD: 0.02; 95% CI: −0.25 to 0.28) and TG (SMD: −0.28; 95% CI: −0.76 to 0.20), compared to control groups. Heterogeneity (*I*^2^) was equal to 79%, 16%, 38%, 0%, and 89%, for TC, LDL-c, HDL-c, LDL-c/HDL-c ratio, and TG, respectively, suggesting low heterogeneity for LDL-c, HDL-c, and LDL-c/HDL-c ratio and high heterogeneity between studies for TC and TG (Figure 8, Figure 9, Figure 10, Figure 11 and Figure 12). Almond (SMD: −1.06; 95% CI: −2.12 to −0.01) and peanut consumption (SMD: −0.63; 95% CI: −1.14 to −0.13) decreased TC compared to controls, with high and low heterogeneity of evidence (*I*^2^ = 90 and 0%), respectively (Appendix A). Moreover, the eating of mixed nuts increased HDL-c compared to controls (SMD: 0.52; 95% CI: 0.11 to 0.92), with low heterogeneity (*I*^2^ = 0%) (Appendix A). In studies with intervention times of less than 12 weeks, nut intake plus an ER diet increased LDL-c (SMD: 0.25; 95% CI: 0.06 to 0.44) and HDL-c (SMD: 0.28; 95% CI: 0.09 to 0.47) compared to controls, with low evidence of heterogeneity (*I*^2^ = 0%) (Appendix A).

In subgroup analysis by nut type and time of intervention, eating nuts had no effects on lipid metabolism markers during ER dietary interventions (Supplementary Appendix A and S18–S21).

#### 3.4.3. Effects of Nut Consumption on Glucose Metabolism during ER Dietary Intervention

General analyses indicated that nut consumption allied to an ER diet had no effects on glucose (SMD: 0.14; 95% CI: −0.16 to 0.44), insulin (SMD: −0.01; 95% CI: −0.21 to 0.20), and HOMA-IR (SMD: 0.12; 95% CI: −0.11 to 0.34), compared to controls. Heterogeneity (*I*^2^) was equal to 71%, 19%, and 0%, for glucose, insulin, and HOMA-IR, respectively, suggesting low heterogeneity for insulin and HOMA-IR, and high heterogeneity between studies for glucose (Figure 13, Figure 14 and Figure 15).

Eating peanuts increased glucose (SMD: 0.31; 95% CI: 0.04 to 0.58) and insulin (SMD: 0.29; 95% CI: 0.02 to 0.56), compared to controls, with low evidence of heterogeneity (*I*^2^ = 0%) (Appendix A). In studies with intervention times of less than 12 weeks, the nut intake combined with an ER dietary intervention increased HOMA-IR (SMD: 0.31; 95% CI: 0.06 to 0.56), compared to controls, with low evidence of heterogeneity (*I*^2^ = 0%) (Appendix A). In subgroup analysis by nut type and time of intervention, eating nuts had no effects on glucose metabolism markers during ER dietary intervention, either (Appendix A–S26).

#### 3.4.4. Effects of Nut Consumption on Blood Pressure during ER Dietary Intervention

According to the main analysis, the intake of nuts allied to an ER diet had no effects on SBP (SMD: −0.07; 95% CI: −0.37 to 0.24) and DBP (SMD: 0.06; 95% CI: −0.13 to 0.26) compared to controls. This suggests low heterogeneity for DBP, and moderate heterogeneity between studies for SBP (Figure 16 and Figure 17).

In subgroup analysis by type of nuts and time of intervention, eating nuts and energy restriction had no effects on blood pressure (Supplementary Appendix A).

## 4. Discussion

The increased consumption of nuts by people with obesity aiming at weight loss justifies the development of this systematic review. To our knowledge, this is the first study to review the effects of nut intake with ER dietary intervention on weight loss, anthropometric and body composition variables, and all traditional cardiometabolic risk factors. In this way, the results showed great heterogeneity in the protocols, including the type of nuts and intervention time. The data from the meta-analysis indicate that nut consumption plus ER did not affect anthropometric and body composition measurement, lipid and glucose metabolism markers, and blood pressure. In subgroup analysis, almond intake decreased WC, almonds and peanuts decreased TC, and mixed nuts consumption increased HDL-c, all compared to controls. In this sense, we believe that the results of the meta-analysis could be different if there were more studies for each type of nut, which present distinct food matrices and corroborate the positive findings.

Based on the data, nuts seem to be an alternative to reduce WC and cardiometabolic risk despite the still incipient results. Waist circumference is the most used method in the literature to assess visceral adiposity, with some cut-off points associated with greater cardiovascular risk and diagnostic criteria for metabolic syndrome [49,50,51,52]. Human studies are controversial regarding the action mechanism of nuts on visceral adiposity and WC. Garrido-Miguel et al. (2021) argues the possibility that components such as muscle strength and cardiorespiratory fitness may mediate the relationship between nut intake and anthropometric measurements [49]. These variables were little considered in the studies analyzed by this systematic review.

Another important point is the ER dietary treatment, a criterion for inclusion in this study. ER and healthy dietary patterns are the gold standard for treatment for weight loss, low-grade inflammation, and comorbidities associated with obesity [53,54]. Corroborating the literature, an ER diet alone, represented by some control groups, also demonstrated a positive impact on weight loss, body composition, and cardiometabolic markers. However, we expected that also adding nut consumption to energy-restricted protocols could have additional results related to cardiovascular risk.

Although nuts have not demonstrated benefits in controlling traditional cardiometabolic risk markers, the polyphenols, unsaturated fatty acids, phytosterols, and fibers found in nuts have already shown actions on lipids [55]. Ellagitannins are a group of polyphenols that have been associated with positive changes in lipoproteins [56]; unsaturated lipids are associated with LDL-c maintenance [57]; phytosterols, due to their structural similarity to cholesterol, have a lipid-lowering action [58]; and soluble fiber slows gastric emptying, hinders diffusion in the small intestine, and increases the excretion of bile acids [59].

However, the beneficial findings were not replicated in all studies, probably due to differences in the types of nuts, doses by day, and intervention periods used. Although they belong to the same food group, the chemical composition of nuts can vary greatly from one another, especially in relation to the content of bioactive compounds. Furthermore, the intervention period varied greatly, between 4 and 72 weeks, as well as doses of 28 to 84 g, with two studies not offering it in grams, one indicated the percentage of the total caloric value (15%) and the other, units (9 and 18 for each group) per day, differences that may interfere with the effects.

However, the consumption of nuts, related to their nutritional components, has presented some evidence of benefits on other relevant mechanisms mediators of the physiopathology of obesity, such as controlling inflammation and oxidative stress [42,47,60], improving dysbiosis and intestinal permeability [60,61], and modulating hunger and satiety hormones [14,16,17].

Evidence from one narrative review suggests that some specific nuts, such as almonds and walnuts, can favorably modify inflammation, and others, such as Brazil nuts, can favorably influence oxidative stress [60]. A systematic review of clinical trials showed that the chronic consumption of Brazil nuts seemed to be effective in improving antioxidant status and changed some oxidative stress markers by increasing selenium, GPx, and selenoprotein in plasma, serum, whole blood, and erythrocytes [23]. Work by Ghanavati et al. (2021) also showed a reduction in inflammatory factors, such as ICAM-1 and IL-6, with intervention with equal amounts of unsalted roasted pistachios, almonds, and peanuts in individuals with overweight or obesity and stable coronary artery disease [42].

Furthermore, the “Brazilian Nuts Study” has presented some additional benefits of Brazil nuts and cashew nuts on endothelial function, inflammation, intestinal microbiota, and intestinal permeability, in addition to satiety control during energy-restricted dietary treatment. In this sense, the consumption of mixed nuts (Brazil nuts and cashews) favored the increase in beneficial bacteria and potentially improved pathways associated with the reduction in body fat, in addition to attenuating the increase in intestinal permeability and inflammation, demonstrated by reduced IL-8, after an 8-week obesity dietary treatment [61]; in addition, Brazil nuts and cashew nuts reduced total body fat and at the same time improved the percentage of lean mass and reduced VCAM-1 concentration, improving endothelial function [47] in addition to decreasing ghrelin [16]. Moreover, Silveira et al. (2024) also showed a reduction in inflammation markers (C-reactive protein, tumor necrosis factor, IL-1β, and IL-8) after 8 weeks of consumption of Brazil nuts rich in selenium (8 g of Brazil nuts providing 347.2 μg selenium—Se) plus energy restriction, compared to the nut-free energy-restricted group [62].

This study has some strengths: the methodological rigor, the broad search in the literature, and the design of the included studies (RCTs), which reduced the risk of bias. The work also has some limitations. The different quantities and types of nuts make it difficult to compare and establish a specific dose and type of nut related to these effects. Moreover, most studies have a moderate risk of bias, which may limit the data interpretation. However, this study points out some gaps that still need to be investigated, such as the role of each of the nuts on body composition and cardiometabolic control, as well as their application in longer-term studies.

## 5. Conclusions

Nut consumption did not provide significant additional benefits for weight loss, body composition, or traditional cardiometabolic risk markers among adults with overweight/obesity consuming energy-restricted diets. While subgroup analyses revealed small changes, the limited number of randomized controlled trials available to date have shown considerable heterogeneity in terms of intervention protocols (e.g., duration, type, and dose of nuts) and participant characteristics (e.g., degree of obesity and health status). Given the diverse nutritional profile of nuts, their widespread consumption, and their recognized role in promoting healthy dietary patterns and managing chronic diseases, further research is necessary to clarify their true impact when incorporated into energy-restricted dietary interventions.

## Figures and Tables

**Figure 1 foods-13-03008-f001:**
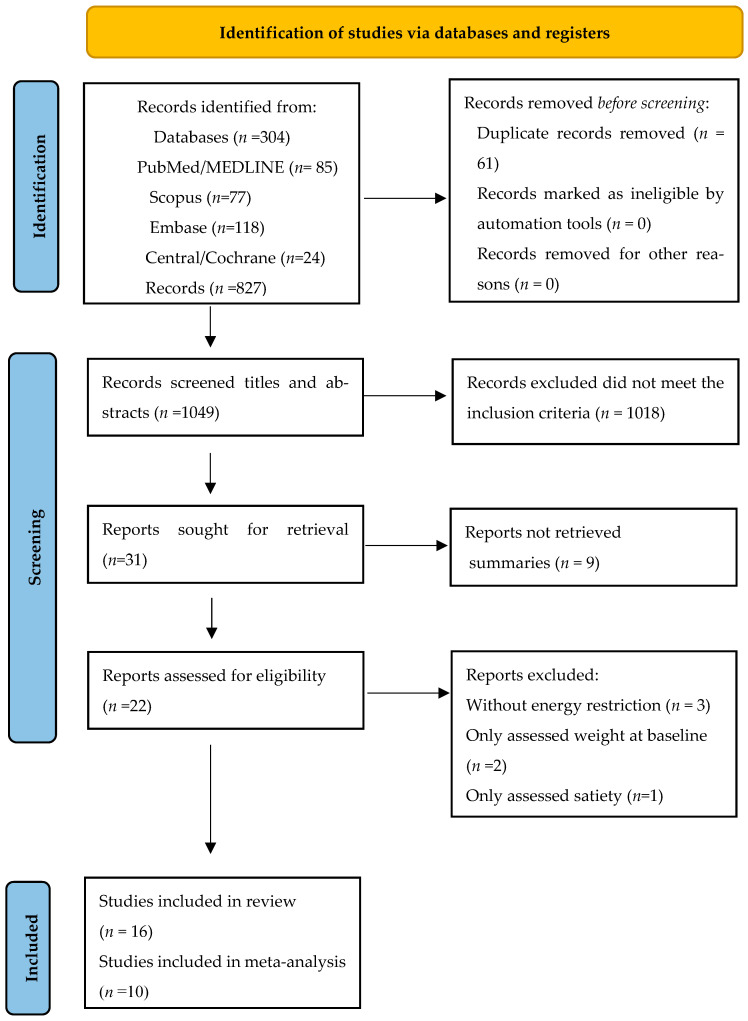
PRISMA 2020 flow diagram for new systematic reviews, which included searches of databases and registers only [29].

**Figure 2 foods-13-03008-f002:**
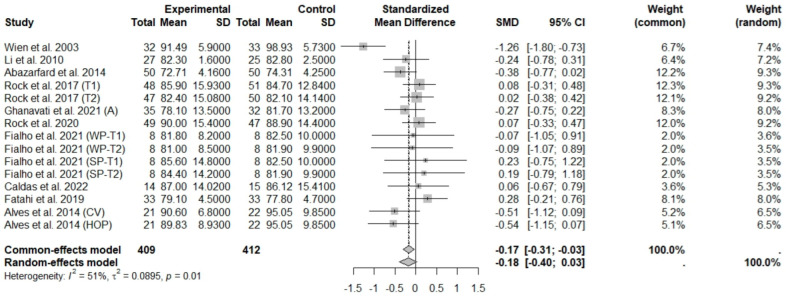
Forest plot of randomized controlled trials investigating the effects of nut consumption combined with energy-restricted diets on weight loss [25,28,37,38,39,42,44,46,47]. Squares represent the weight of studies in meta-analysis, and horizontal lines, the 95% CI; diamond’s center represents the combined treatment effect and horizontal tips represent the 95% CI. CI: confidence interval; SMD: standard mean difference. T1, first assessment after intervention; T2, second assessment after intervention; CV, conventional peanuts group; HOP, high-oleic peanuts group; WP, whole roasted peanuts group; SP, skinned peanuts group.

**Figure 3 foods-13-03008-f003:**
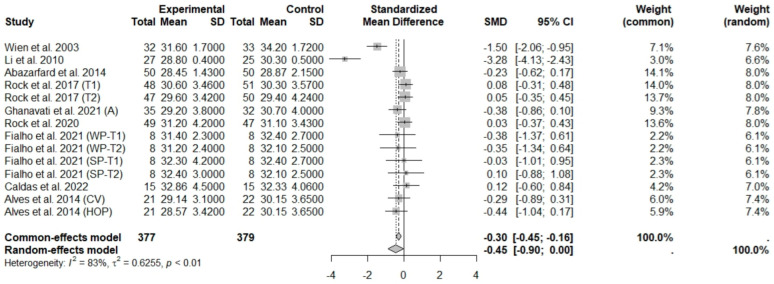
Forest plot of randomized controlled trials investigating the effects of nut consumption combined with energy-restricted diet on body mass index [25,28,37,38,39,42,44,46,47] Squares represent the weight of studies in meta-analysis, and horizontal lines, the 95% CI; diamond’s center represents the combined treatment effect and horizontal tips represent the 95% CI. CI: confidence interval; SMD: standard mean difference. T1, first assessment after intervention; T2, second assessment after intervention; CV, conventional peanuts group; HOP, high-oleic peanuts group; WP, whole roasted peanuts group; SP, skinned peanuts group.

**Figure 4 foods-13-03008-f004:**
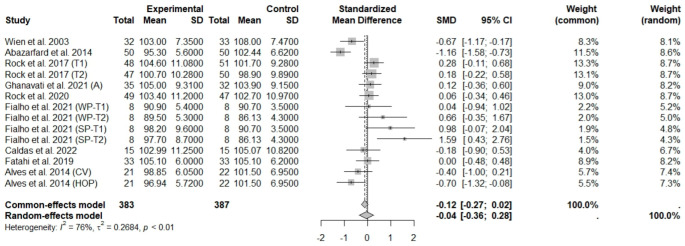
Forest plot of randomized controlled trials investigating the effects of nut consumption combined with energy-restricted diet on waist circumference [25,28,37,39,41,42,44,46,47]. Squares represent the weight of studies in meta-analysis and horizontal lines, the 95% CI; diamond’s center represents the combined treatment effect and horizontal tips represent the 95% CI. CI: confidence interval; SMD: standard mean difference. T1, first assessment after intervention; T2, second assessment after intervention; CV, conventional peanuts group; HOP, high-oleic peanuts group; WP, whole roasted peanuts group; SP, skinned peanuts group.

**Figure 5 foods-13-03008-f005:**
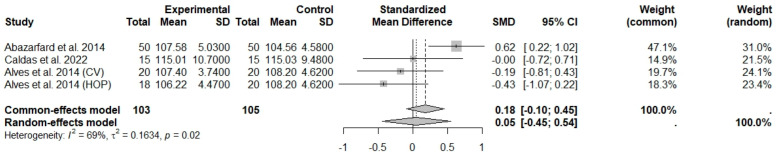
Forest plot of randomized controlled trials investigating the effects of nut consumption combined with energy-restricted diet on hip circumference [25,44,47]. Squares represent the weight of studies in meta-analysis and horizontal lines, the 95% CI; diamond’s center represents the combined treatment effect and horizontal tips represent the 95% CI. CI: confidence interval; SMD: standard mean difference. CV, conventional peanuts group; HOP, high-oleic peanuts group.

**Figure 6 foods-13-03008-f006:**
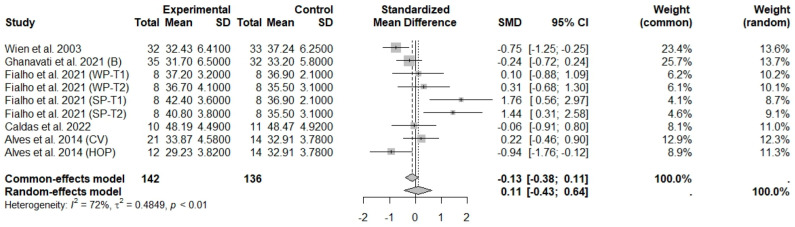
Forest plot of randomized controlled trials investigating the effects of nut consumption combined with energy-restricted diet on fat mass [28,42,44,46,47]. Squares represent the weight of studies in meta-analysis and horizontal lines, the 95% CI; diamond’s center represents the combined treatment effect and horizontal tips represent the 95% CI. CI: confidence interval; SMD: standard mean difference. T1, first assessment after intervention; T2, second assessment after intervention; CV, conventional peanuts group; HOP, high-oleic peanuts group; WP, whole roasted peanuts group; SP, skinned peanuts group.

**Figure 7 foods-13-03008-f007:**
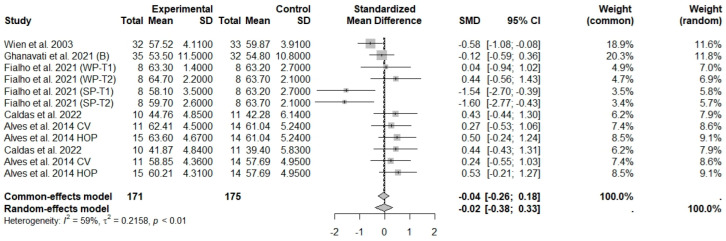
Forest plot of randomized controlled trials investigating the effects of nut consumption combined with energy-restricted diet on fat-free mass or lean mass [28,41,42,44,47]. Squares represent the weight of studies in meta-analysis and horizontal lines, the 95% CI; diamond’s center represents the combined treatment effect and horizontal tips represent the 95% CI. CI: confidence interval; SMD: standard mean difference. T1, first assessment after intervention; T2, second assessment after intervention; CV, conventional peanuts group; HOP, high-oleic peanuts group; WP, whole roasted peanuts group; SP, skinned peanuts group.

**Figure 8 foods-13-03008-f008:**
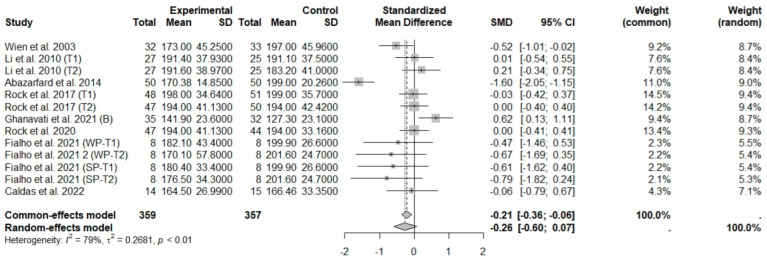
Forest plot of randomized controlled trials investigating the effects of nut consumption combined with energy-restricted diet on total cholesterol [25,28,37,38,39,43,46,47]. Squares represent the weight of studies in meta-analysis and horizontal lines, the 95% CI; diamond’s center represents the combined treatment effect and horizontal tips represent the 95% CI. CI: confidence interval; SMD: standard mean difference. T1, first assessment after intervention; T2, second assessment after intervention; WP, whole roasted peanuts group; SP, skinned peanuts group.

**Figure 9 foods-13-03008-f009:**
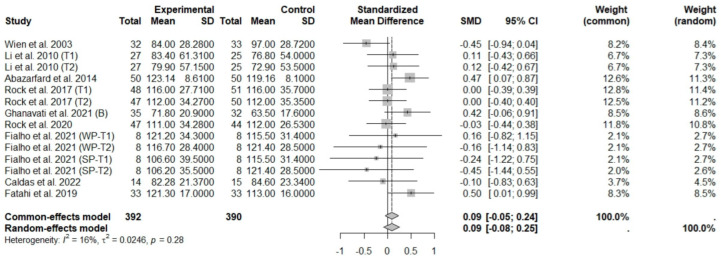
Forest plot of randomized controlled trials investigating the effects of nut consumption combined with energy-restricted diet on low-density lipoprotein cholesterol [25,28,37,38,39,41,43,46,47]. Squares represent the weight of studies in meta-analysis and horizontal lines, the 95% CI; diamond’s center represents the combined treatment effect and horizontal tips represent the 95% CI. CI: confidence interval; SMD: standard mean difference. T1, first assessment after intervention; T2, second assessment after intervention; WP, whole roasted peanuts group; SP, skinned peanuts group.

**Figure 10 foods-13-03008-f010:**
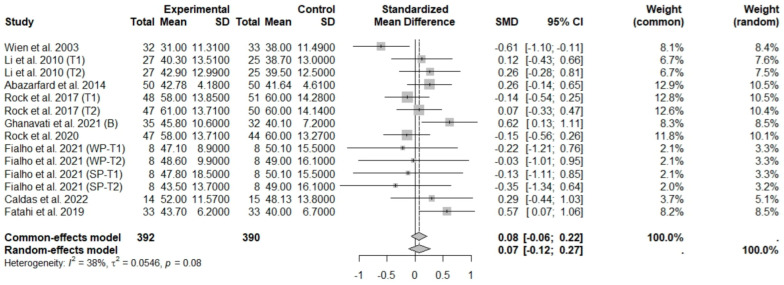
Forest plot of randomized controlled trials investigating the effects of nut consumption combined with energy-restricted diet on high-density lipoprotein cholesterol [25,28,37,38,39,41,43,46,47]. Squares represent the weight of studies in meta-analysis and horizontal lines, the 95% CI; diamond’s center represents the combined treatment effect and horizontal tips represent the 95% CI. CI: confidence interval; SMD: standard mean difference. T1, first assessment after intervention; T2, second assessment after intervention; WP, whole roasted peanuts group; SP, skinned peanuts group.

**Figure 11 foods-13-03008-f011:**
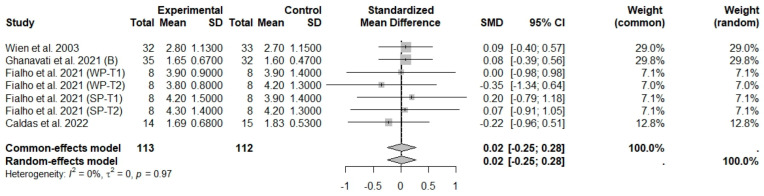
Forest plot of randomized controlled trials investigating the effects of nut consumption combined with energy-restricted diet on low-density lipoprotein cholesterol/high-density lipoprotein cholesterol ratio (LDL-c/HDL-c ratio) [28,43,46,47]. Squares represent the weight of studies in meta-analysis and horizontal lines, the 95% CI; diamond’s center represents the combined treatment effect and horizontal tips represent the 95% CI. CI: confidence interval; SMD: standard mean difference. T1, first assessment after intervention; T2, second assessment after intervention; WP, whole roasted peanuts group; SP, skinned peanuts group.

**Figure 12 foods-13-03008-f012:**
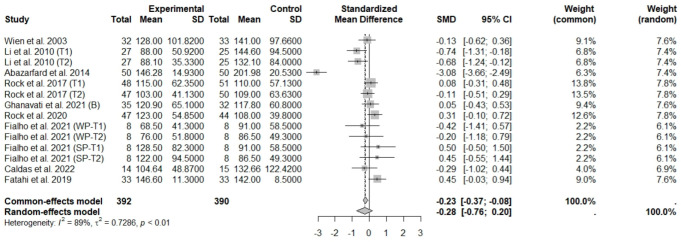
Forest plot of randomized controlled trials investigating the effects of nut consumption combined with energy-restricted diet on triglyceride levels [25,28,37,38,39,41,43,46,47] Squares represent the weight of studies in meta-analysis and horizontal lines, the 95% CI; diamond’s center represents the combined treatment effect and horizontal tips represent the 95% CI. CI: confidence interval; SMD: standard mean difference. T1, first assessment after intervention; T2, second assessment after intervention; WP, whole roasted peanuts group; SP, skinned peanuts group.

**Figure 13 foods-13-03008-f013:**
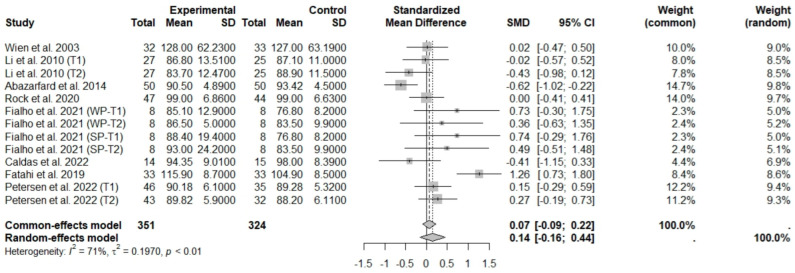
Forest plot of randomized controlled trials investigating the effects of nut consumption combined with energy-restricted diet on glycaemia [25,28,38,39,41,46,47,48]. Squares represent the weight of studies in meta-analysis and horizontal lines, the 95% CI; diamond’s center represents the combined treatment effect and horizontal tips represent the 95% CI. CI: confidence interval; SMD: standard mean difference. T1, first assessment after intervention; T2, second assessment after intervention; WP, whole roasted peanuts group; SP, skinned peanuts group.

**Figure 14 foods-13-03008-f014:**
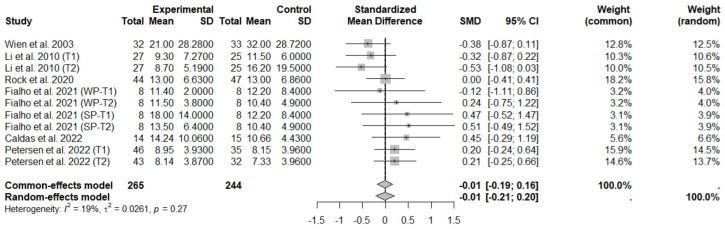
Forest plot of randomized controlled trials investigating the effects of nut consumption combined with energy-restricted diet on insulinemia [28,38,39,46,47,48]. Squares represent the weight of studies in meta-analysis and horizontal lines, the 95% CI; diamond’s center represents the combined treatment effect and horizontal tips represent the 95% CI. CI: confidence interval; SMD: standard mean difference. T1, first assessment after intervention; T2, second assessment after intervention; WP, whole roasted peanuts group; SP, skinned peanuts group.

**Figure 15 foods-13-03008-f015:**
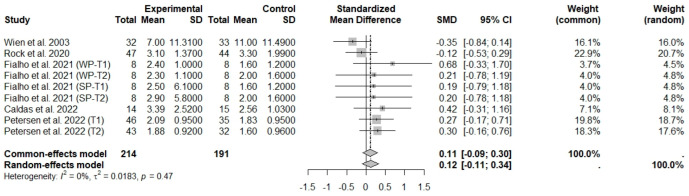
Forest plot of randomized controlled trials investigating the effects of nut consumption combined with energy-restricted diet on homeostatic model assessment for insulin resistance (HOMA-IR) [28,39,46,47,48]. Squares represent the weight of studies in meta-analysis and horizontal lines their 95% CI; diamond’s center represents the combined treatment effect and horizontal tips represent the 95% CI. CI: confidence interval; SMD: standard mean difference. T1, first assessment after intervention; T2, second assessment after intervention; WP, whole roasted peanuts group; SP, skinned peanuts group.

**Figure 16 foods-13-03008-f016:**
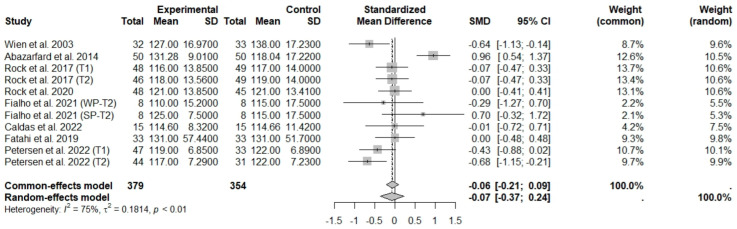
Forest plot of randomized controlled trials investigating the effects of nut consumption combined with energy-restricted diet on systolic blood pressure [25,28,37,39,41,46,47,48]. Squares represent the weight of studies in meta-analysis and horizontal lines, the 95% CI; diamond’s center represents the combined treatment effect and horizontal tips represent the 95% CI. CI: confidence interval; SMD: standard mean difference. T1, first assessment after intervention; T2, second assessment after intervention; WP, whole roasted peanuts group; SP, skinned peanuts group.

**Figure 17 foods-13-03008-f017:**
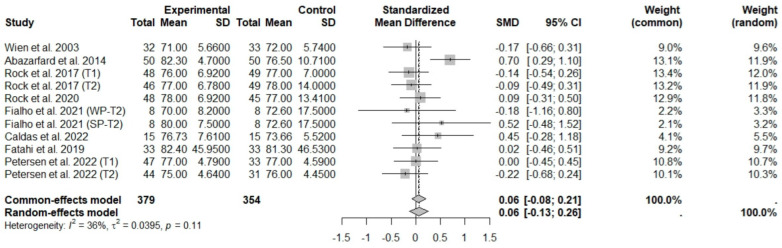
Forest plot of randomized controlled trials investigating the effects of nut consumption combined with energy-restricted diet on diastolic blood pressure [25,28,37,39,41,46,47,48]. Squares represent the weight of studies in meta-analysis and horizontal lines, the 95% CI; diamond’s center represents the combined treatment effect and horizontal tips represent the 95% CI. CI: confidence interval; SMD: standard mean difference. T1, first assessment after intervention; T2, second assessment after intervention; WP, whole roasted peanuts group; SP, skinned peanuts group.

## Data Availability

The raw data supporting the conclusions of this article will be made available by the authors on request.

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
