# Peer review of "Effect of Nuts Combined with Energy Restriction on the Obesity Treatment: A Systematic Review and Meta-Analysis of Randomized Controlled Trials"

_foods, 2024, doi:10.3390/foods13183008_

Round 1
Reviewer 1 Report
Comments and Suggestions for Authors
Thank you very much for performing this well designed and written manuscript. There are minor comments that I would like from you to addressed:
1. Can you include results of meta analysis in abstract.
2. Please mention how did you conduct meta analysis in methods.
3. Mention results of assessment of risk of bias in results.
Author Response
Thank you for nice comment, we made changes in the revised manuscript as suggested and highlight in red color.
- Can you include results of meta analysis in abstract.
Thank you for your comment, the methods and results of the meta-analysis were included in the revised abstract (lines 21-22 and 28-29).
- Please mention how did you conduct meta analysis in methods.
For some mistake, we did not present this information. Now, the revised manuscript contains the correct description of meta-analysis methods (lines 170-185).
- Mention results of assessment of risk of bias in results.
Thank you for your comment. The results of the bias analysis are included in the results (lines 412-427) and have been highlighted in red for reference purposes only.
Reviewer 2 Report
Comments and Suggestions for Authors
Title: Should be reduced, it is too long and should not have the results.
Introduction: in lines 47 to 63, it is necessary to complement the amounts or values of the doses at which those potential benefits listed are obtained.
It is a well-conducted study; it is an exhaustive work in analyzing and presenting results. The discussion and conclusions are thoughtful, well-written, and reflect the most relevant results.
Author Response
We appreciate the reviewer’s comments, which are addressed in the revised manuscript (highlight in red color).
- Title: Should be reduced, it is too long and should not have the results.
We agree with the reviewer, the title has been shortened and the results have been removed, as suggested.
- Introduction: in lines 47 to 63, it is necessary to complement the amounts or values of the doses at which those potential benefits listed are obtained.
We added suggested information in the Introduction section (Lines 47 to 63), as suggested.
- It is a well-conducted study; it is an exhaustive work in analyzing and presenting results. The discussion and conclusions are thoughtful, well-written, and reflect the most relevant results.
We appreciate your comment, we strive to ensure that the quality of the text meets the requirements of the journal's reviewers.
Reviewer 3 Report
Comments and Suggestions for Authors
Dear Authors,
I thank the Editor for entrusting me to review this manuscript.
Nuts are an important part of the human diet. Depending on their type, nuts are a valuable source of protein, fibre, unsaturated fatty acids, folic acid and many vitamins. Nutritionists recommend their regular consumption. So, kudos to the authors of this manuscript for undertaking a review of the literature in this area in the context of weight loss and cardiometabolic diseases and synthesising the information obtained in the form of a meta-analysis.
Below are my suggestions / comments:
Using a consecutive selection method, the authors used 16 published scientific articles available in various databases. The results from the analysed papers were compiled and recorded in a readable form in the summary in Table 1.
The meta-analysis was conducted using fewer publications due to the lack of specific information. The meta-analyses were conducted in detail and the results, despite their multiplicity, were presented clearly.
Author Response
We appreciate your comments. Nuts really have high nutritional value due to their rich nutritional composition containing bioactive macro and micronutrient compounds relevant to human health. Our research group has dedicated itself to extensively studying these foods, especially Brazilian nuts (cashew nuts and Brazil nuts), and has found good results. We appreciate your compliments and interest in our work.